# Identifying the Black Country’s Top Mental Health Research Priorities Using a Collaborative Workshop Approach: Community Connexions

**DOI:** 10.3390/healthcare12242506

**Published:** 2024-12-11

**Authors:** Hana Morrissey, Celine Benoit, Patrick Anthony Ball, Hannah Ackom-Mensah

**Affiliations:** 1Consultant Pharmacist, Perth, WA 6036, Australia; 2School of Sociology and Social Policy, Law and Social Sciences Building, University Park, Nottingham NG7 2RD, UK; celine.benoit@nottingham.ac.uk; 3Birmingham Community Healthcare NHS Foundation Trust (BCHC), Birmingham B4 7ET, UK; 4School of Medical and Dental Sciences, Charles Sturt University, Wagga, NSW 2678, Australia; pba87218@gmail.com; 5Underserved Populations Community Development & Community Connexions, Black Country Healthcare NHS Foundation Trust (BCHFT), Research and Innovation, Dudley DY2 8PS, UK; hannah.ackom-mensah@nhs.net

**Keywords:** inequality in mental health, access to mental health services, self-stigmatisation, underserved communities, cultural disparity

## Abstract

**Background:** The Black Country (BC) is an area of the United Kingdom covering Dudley, Sandwell, Walsall, and Wolverhampton. The area is ethnically, culturally and religiously diverse. One-fifth of the total population is in the lowest socioeconomic quintile, with an uneven distribution of wealth. The area manifests unmet needs and as perceived underserved community groups. **Objectives and Methods:** To better understand the situation and inform future provision, listening events were organised across the BC to engage with local underserved communities. A mixed-methods design was employed, using collaborative workshops. The workshops enabled stakeholders to explore priorities, perceived barriers and solutions to mental health services’ access within the BC. **Results:** Sixty participants verbally consented and signed in to attend the three workshops. There were nine groups that provided 247 statements on the topic, yielding a total of 12 codes and six themes (priorities). The top identified priorities were inappropriate periodisation of accessible funded healthcare needs (n = 42, 18.03%), barriers to appropriate healthcare (n = 49, 21.03%) and limited resources for training, health promotion, preventative care and support networks (n = 62, 26.61%). **Conclusions:** Addressing the identified priorities will require location and community-specific solutions to establish those communities’ trust and engagement. Cultural stigma should not be viewed as the only barrier to access healthcare but should be considered in combination with the population’s reluctance to reach out to healthcare services due to loss of trust between community groups and lack of co-design of culturally and religiously appropriate services for the community.

## 1. Background

In the United Kingdom (UK), the ‘Black Country’ refers to an area of the West Midlands that comprises four Local Authorities: Dudley, Sandwell, Walsall and Wolverhampton. The Black Country was part of the birth of the Industrial Revolution [1], and its name is thought to derive either from the abundance of coal in the area, or the pollution that arose from its use to fuel the industrial activity. The precise boundaries have changed over the years [1,2], but in terms of health management, since 1 July 2022, the area has been managed by an Integrated Care Board (ICB), which was tasked to manage health and social care services for the Black Country [3]. The ICB set out its vision in the NHS Black Country Joint Forward Plan 2023–2028, updated in April 2024 [4].

In 1785, the route across this area from the community that later became the City of Birmingham to Wolverhampton was described as ‘one continuous town’ [2]. The major road following this route, completed in 1927, still runs through continuous conurbation from end to end. However, what appears to be a mass of uniform urban sprawl comprises many local communities with strong historic identities arising from their industrial past, such as chain making in Netherton or leather making in Walsall. The early Industrial Revolution in the UK was characterised by poor employment conditions, poor housing and high levels of industrial pollution of the air, land and water, like the ‘dark Satanic Mills’ of William Blake’s 1804 poem [5]. By the 1960s and 1970s, the area had good working conditions, above-average wages and low unemployment. During the growth years, supplying the workforce for the industries brought many immigrants to the area, most of whom settled and became part of the community, bringing their cultures and religions [6] and making the Black Country ‘super diverse’ [7]. However, between 1979 and 1982, most of the major employers closed and their manufacturing moved overseas, creating large-scale unemployment and massive de-industrialisation, and this is still reflected in the health landscape today [6].

The local government Black Country Consortium published the Black Country & West Birmingham Socio-Economic profile in July 2020 [6] that presented the population demographics and compared the infrastructure and the health and education outcomes of the Black Country to the wider UK. It reported relatively more children aged 0–15 (21.5%) than the national average (19.2%). In terms of the aging population, Sandwell, Walsall and Wolverhampton are slightly below the average, whilst Dudley is slightly above.

The region is multi-cultural. The population identifying as Asian, Black or other minority ethnic groups is above the national average of 14% in Sandwell (40%), Wolverhampton (39%) and Walsall (28%), whilst Dudley (14%) has the national average.

The 2023 forward plan [4] provides some statistics that establish disparities in outcomes, including in mental health, such as obesity with 69% of adults identified (National average 64%), alcohol-related conditions accounted for 505 hospital admissions per 100,000 (National 456 per 100,000) and premature mortality in severe mental illness registered 117 per 100,000 vs. 104 per 100,000 nationally [4]. However, such ‘community snapshot’ data lack the resolution to drill down to individual communities, origins and backgrounds. This study was designed to develop links and pathways to these groups within the community in order to engage with them and to understand where and how services fail to address their needs.

The Black Country ICS population has been recognised as the second most deprived nationally. At the national level, 20% of the population is classified as in the most deprived quintile, but Dudley (28%), Walsall and Wolverhampton (both 52%) and Sandwell (60%) are well above the national average. This is reflected in employment, income, home and vehicle ownership and air quality [6]. These underserved groups also have generally worse mental health outcomes [8]. In any particular health service catchment area, there is likely to be significant diversity that services must attempt to adequately address whilst observing cultural and religious differences, and community perceptions of certain illnesses; particularly issues such as mental illness [9,10].

The aim of this project was to involve underserved communities to identify their community’s mental health service priorities. The number one priority was to improve access and quality of services. Previous research undertaken during the COVID pandemic has shown differences in the uptake of services by particular groups in the community [11,12,13]. To progress this priority, it is essential to build open and effective communication links with the communities served so that providers can fully understand how best to tailor their services to effectively reach all who need them.

This project stems from ‘Community Connexions’ [14], a patient and public engagement programme led by the Black Country Healthcare (NHS) Foundation Trust (BCHFT) [3] and the Birmingham Community Healthcare NHS Foundation Trust (BCHC) [15]. Aston University was a key academic partner. By engaging with underserved communities, sharing understandings, building trust, and actively listening, they collaborated with various community, voluntary sector, faith-based and social enterprise (VCFSE) organisations, academics, local councillors and other stakeholders. This approach was adopted to understand the health needs and priorities of local communities, especially the underserved populations, their health behaviours and the barriers leading to poor engagement with health services, and to research and disseminate findings on this.

Over two years, listening events were organised across the Black Country and Birmingham to offer opportunities to engage with local communities, aiming especially to engage underserved populations. This paper focuses on the data collected in the Black Country. A frequent enquiry from the communities was: What are the topmost mental health priorities in the region, and how are they being addressed? Community forums, public meetings, and local community events provided avenues to enable local communities to voice their mental health concerns [16].

The Community Connexions Operational Group is an advisory body for Community Connexions within the Black Country, made up of academics, a voluntary and community-sector representative, an NHS Trust community mental health representative, a consultant with local government expertise and a representative from the Clinical Research Network, West Midlands. To commence the work in these community groups, it was important to review historical data and projects. There were a great deal of personal experiences. However, this was not recorded or acknowledged as official data. As such, it was important to understand the local population in order to be able to better serve them, with a specific focus on underserved communities.

The project idea was conceptualised to understand the reasons behind the apparent distrust and low participation of some local communities in clinical research and to build a network of individual experts, organisations and those with an interest in tackling mental health inequalities and enhancing the quality of life. The most logical starting point was listening to the local communities’ priorities and their experiences in accessing services and participating in research. Additionally, it was important to have community members, practitioners, local government, non-government organisations, faith leaders, community leaders, academics and researchers in the same room, so that they could share their views in an open safe place. In collaboration with this operational group, the Community Connexions–Community Health Co-Production Forum was planned and hosted in March 2024 to identify the top three mental health priorities across the region. This was a participatory networking opportunity to raise awareness of all the organisations working across the Black Country directly or indirectly supporting mental health and to co-produce and share the outcomes through dissemination methods such as academic and conference publications, publications in local newspapers, posters, poetry and blogs. A better understanding of the priorities and challenges facing Black Country communities enables researchers and service providers to plan effective interventions and support. However, there is a lack of community-driven data [17].

## 2. Aim

This paper outlines the methodological approach and outcomes of the first BCHFT Community Connexions–Community Health Co-Production Forum hosted in the Black Country. The aim was to understand the Black Country community stakeholders’ top 3 mental health priorities and suggest solutions to be the focus of future research and collaborative working across the Black Country [4].

## 3. Objectives

Identifying the Black Country’s top 3 mental health priorities (WS-1);Utilising research to explore solutions for the identified issues (WS-2);Identifying the next step approach (WS-3).

## 4. Setting

The research was located in the Black Country, in the UK West Midlands. A significant number of the communities originate from Middle Eastern, African, Caribbean and South Asian heritage, with other minoritised groups also present. The region’s cultural diversity, high level of deprivation, health disparities, socioeconomic challenges, and high rate of unemployment disproportionately affect underserved communities and minoritised ethnic groups. These are some of the factors influencing the mental health of our communities. The Black Country Integrated Care Systems (ICS) area has one of the highest deprivation levels in England [6,18]. Providing an understanding of this context is important to identify and address vital and impactful mental health issues in our communities, especially as the Black Country Integrated Care Board (ICB) views support for people with severe mental illness as care primarily delivered in the community [19].

## 5. Methods

The participatory, unstructured focus group approach adopted was intended to interactively gather perspectives from groups assembled by the researchers from a large number of diverse participants. The focus group was selected based on two underpinning theories: the Sociocultural learning theory and the Participatory research theory. The Sociocultural theory emphasises that “learning is a social process, within the individual’s sociocultural context, which continually influences our thoughts, actions, and perceptions of ourselves as well as of the world around us” [20].

Participatory research is primarily centered on the participant's voice and experiences [21]. Combining elements of these theories, we created an atmosphere that nurtured conversation among equal participants and encouraged the exchange of ideas. Informed by our community partners, we fostered a dialogical tone that encouraged participants to share, reflect on, and collaboratively make meaning from their everyday lived experiences [22].

All participants were informed that data gathered from the workshops would be used by Community Connexions to identify the top 3 mental health priorities in the Black Country. All attendees consented to their anonymised data being collected for research and publishing purposes.

A mixed-methods design was employed. The mode of inquiry was qualitative, collaborative workshops [12,23] and quantitative evaluation questionnaires to foreground community members’ voices. The workshops enabled all stakeholders to explore priorities, (perceived) barriers and solutions to mental health within the Black Country. The final workshop and quantitative evaluation provided insights into the community’s perceptions about the next steps for BCHFT Community Connexions.

The participants were immersed in a series of three episodes of collaborative workshops held over 3 h on a single day. Each step was designed to sequentially build on a previous workshop to explore stakeholder perceptions of mental health priorities and the potential solutions. At each sub-group table, detailed and nuanced understandings and challenges offered in narrative format were captured in a notational format using single words and simple sentences. This was an approach to ensure that the stakeholders’ and underserved populations’ needs and aspirations were aligned with the identified mental health priorities. The collaborative methodology built trust and provided opportunities for all community members to be involved as active participants in matters of concern to them.

## 6. Participants

Volunteer, snowball, and purposive sampling were employed to recruit a varied representation of 60 diverse individuals from the Black Country. The participants included people with lived experiences [16] of poor mental health, VCFSE organisations, academics, healthcare providers, representatives of the National Health Service (NHS) Trusts and local authorities. The selection criteria considered participants comprising diverse representatives from a demographic sample of ages, genders, ethnic and faith backgrounds, socioeconomic status and varied experience with mental health, including gender-specific lived-experience mental health and non-binary individuals and service users from diverse mental health conditions. There was further representation from mental health practitioners and community leaders. Additionally, there was representation from underserved communities, including migrants, asylum seekers, refugees, people who identified as LGBTQ+, neurodivergent, visually impaired, and/or the disabled [24]. This ensured that a wide breadth of mental health priorities and solutions were explored. The demographic diversity was to incorporate differing perspectives on mental health and how economic factors influence mental health priorities and access to mental health.

## 7. Data Collection Tools, Instruments and Procedures

Mixed methods were employed to collect data. Initially, qualitative methods were used to tease out the mental health priorities of the participants and the solutions perceived. This was sequentially followed by quantitative methods (survey) to evaluate the research process and collect data on future engagement with our local communities.

Sixty participants engaged in three episodes of collaborative workshops in a chosen space over 3 h on a single day. Each participant randomly joined 9 out of 10 set-up tables at different times with material laid out to enable informed participatory approaches. These included the agenda for the day, 3 × A1 sheets, 2 × A3 (Flip chart sheets), Post-it note pads, markers, pens, and small plain stickers (used as participant identification name tags), and a copy of the “Community Connexions engagement handbook: Engaging with underserved communities” [23]. This was a way to initiate dialogue and offered all participants conversation starters.

The facilitator for the forum and Community Connexions Lead presented a main question for Workshops 1 and 2 and suggested four supplementary questions designed around the main question to keep the dialogue ongoing at each table about the main issue being explored. The workshops were interspersed with short presentations about Community Connexions, mental health, well-being, co-production, community inclusion and a lunch break. One person acted as a timekeeper to ensure that the event was on schedule. Four other facilitators rotated and randomly joined table discussions during the workshops. One of the facilitators collected scribed sheets from each table after each workshop.

At the analysis and report writing stage, all community representatives were approached with an update about the progress being made to identify their top three mental health priorities. Peripheral activities included a showcase of stands for four local community organisations, including Community Connexions/ BCHFT Research and Innovation. There was also a display corner with posters about Community Connexions community engagements across Black Country and Birmingham. All participants were offered opportunities to write or mark-make their top three mental health priorities and comments. Three excerpts from emails related to this were displayed in this space.

## 8. Workshops

***Workshop 1***: Identifying our top three mental health priorities

In this episode, participants were randomly seated around the allocated tables to facilitate discussions and activities for 30 min in an inclusive atmosphere. The dialogue was open, and all were provided opportunities to explore, question, voice and share their thoughts and perceptions about mental health priorities. The main question was shared on a big screen and sub-questions (SQ) enabled conversations around the main question (MQ). Sub-questions (SQ) are listed in Box 1.

Box 1What are the top three mental health priorities in the Black Country?
SQ1: What mental health challenges do you or people around you face on a regular basis?SQ2: In your opinion, what are the top 3 priorities for improving mental health support in our local communities?SQ3: What resources or support do you feel are lacking for addressing mental health needs?SQ4: How accessible are mental health services for individuals in underserved populations?Qualitative methods enabled the capturing of a wide range of experiences.


***Workshop 2***: Explore solutions on how to impact our communities—Research?

The second episode focused on collaboratively reviewing present solutions or strategies and exploring and making recommendations to address the identified mental health priorities. For 20 min, participants discussed and documented their thoughts and perceptions on research roles in solving mental health issues in underserved populations through community interventions or other changes and solutions that provide positive mental health and well-being outcomes.

The main question (MQ) and sub-questions (SQs) shared on a big screen enabled conversations around the main question. SQs are listed in Box 2.

Box 2Explore solutions on how to impact our communities—Research?
SQ1: What specific challenges do underserved populations face when they need to access mental health support?SQ2: What resources are currently available, and how can we make them more accessible and visible to those who need them most?SQ3: How can we collaborate with local organisations and community leaders to raise awareness and reduce stigma surrounding mental health?SQ4: What innovative approaches or strategies can we implement to ensure long-term sustainability and effectiveness in addressing these mental health priorities?


***Workshop 3***: Community Connexions—What next?

The final episode was a 10-minute discussion to suggest an action plan for the next stage of a Community Connexions community engagement strategy to maintain the momentum generated by the dialogue at the Community Health Co-Production Forum.

## 9. Results

Sixty participants signed up for the event and attended the three workshops. There were nine groups that provided 247 statements on the topic. There were a total of 12 codes generated and agreed on by the researchers for Workshops 1 and 2. Codes were repeated 859 times, with the most repeated code being code 9 (17.46%) and the least repeated code being code 12 (1.98%). Codes 8 and 9 were mentioned by all groups in the workshops (Table 1), and 14 statements were collected at Workshop 3 (Figure 1). A total of 10 themes were generated. The most repeated theme in Workshop 1 was Theme 3 (31%), and the most repeated theme in Workshop 2 was Theme 4 (9.44%). The top three priorities were limited resources for training, health promotion, preventative care and support networks (31%), barriers to appropriate healthcare (25%) and research on new health services model (24%) (Table 2).

At the end of Workshop 3, participants provided their feedback (n = 37 forms). Questions 1 and 2 were related to rating the community event and its content (Figure 2), where most participants rated the event as good and the content as excellent.

When the participants’ responses to Question 3 were analysed (Figure 3), the most reported codes were the group collective experience and the ability to establish a collaboration network.

For Question 4 (Figure 4), the most repeated code was increasing networking time.

Question 5 (Figure 5), the most repeated code was also more time and frequencies of networking events.

## 10. Discussion

Globally, mental health provision follows one of three models: private (user pays), charitable (donor pays) and public (state-funded, free or cost-limited co-payment at point of care). Public systems such as in the UK tend to be funded on capitation models, based upon large-scale census data with some adjustment for identified areas of special need [25]. The Black Country population differs significantly from UK averages in a number of respects. Further, UK healthcare funding from 2010 to 2020 grew at a slower rate than both general and health-related inflation [26].

Patients have traditionally self-presented to their providers or had been referred as a consequence of some type of crisis. Service provision has been supply-side limited, with waiting times recently rising from months to even years to see a specialist for a definitive diagnosis and treatment plan. The only ways around this are through the ability to pay privately for treatment or through charitable provision, which, being even more cash-limited than the public service, will similarly attempt to prioritise those in greatest need. This leads to an environment of services fire-fighting crises instead of early identification of needs before serious problems develop. Mental health is no different from systemic disease in that, in many cases, if detected early and appropriately managed, it is better for the patient and cheaper for society.

This study demonstrates clear and widespread dissatisfaction across all aspects of the current service. The impetus to find new approaches arises from the existing shortfall in services; the growing identification of mental health co-morbidity in chronic systemic diseases, such as heart failure [27], Diabetes [28] and rheumatoid disease, and its negative effect on prognosis; the availability of validated screening tools (examples include CUXOS for anxiety [29] and CUDOS for depression [30]) for screening patients to find those at high risk and intervene much earlier; plus the growing evidence of a failure to reach particular parts of the community [31]. Such a new approach is unlikely to look much like the present failing system, but with new tools to facilitate doing things differently and a better understanding of the needs of the populations to be served, what they desire, and what they perceive they need [11,13], it should be possible to design something that works better.

Bansal et al. [32] investigated ethnic inequalities in mental health and explored the sources. Current service provision discourages many minoritised ethnic populations because of what they described as a ‘monocultural framework of assessment and treatment’. Many in these groups perceive their condition as arising from traumatic experiences during migration and/or experiences of racism. Negative experiences lead to avoidance. The day was designed to include four presentations (Community Connexions in Black Country, Mental Health in the Black Country Community Inclusion Workers and Co-production) and three workshops (Identifying our top 3 Mental Health Priorities, Exploring solutions on how to impact our communities through Research participation, and Community Connexions: next step discussion). The top identified priorities for research topics that may contribute to improving mental health in the Black Country area were inappropriate periodisation of accessible funded healthcare needs (n = 42, 18.03%), barriers to appropriate healthcare (n = 49, 21.03%) and limited resources for training, health promotion, preventative care and support network (n = 62, 26.61%). However, as noted by Winsper [33], whilst priorities may appear very similar, addressing them will require site-specific, and even within a particular location, community-specific solutions. This will require the establishment of enduring links, often in the face of limited engagement or trust [33].

It was clear during the workshops that the participants were very keen to understand the variable communities’ cultures, religions and social hierarchies in a safe space away from formalities or competitions. Researchers and stakeholders were as keen as the members of the communities to voice their interest in understanding community needs to inform their plans to improve services, where researchers and community leaders were keen to team up and collaborate in co-designing and co-producing future research. It is clear from the data here that, although there has been a great deal of discussion about underserved communities avoiding services they do not deem appropriate, and this remains a hugely important area to address, the clearest barrier to access is currently insufficient service availability.

The development of integrated care pathways within the NHS is one strategy aimed at facilitating the identification and realisation of their goals. Building links and establishing trust facilitates the evolution of solutions tailored to specific situations, and community involvement also builds a greater understanding of the realities of healthcare at the management level in terms of how priorities can be addressed within resource limitations [33]. There has to be recognition that resources are limited, and compromises will be required, but these are more likely to be successful and enduring when mutually agreed in partnership. This finding further emphasises the fact that we need to stop putting the onus on the individual (as not obtaining support because of cultural stigma) and shift this to the systems which are failing underserved communities that are actually trying to reach out to them. In particular, providing services that underserved communities are comfortable reaching out to rather than blaming people for not accepting the standard offering is essential.

Winsper [33] also noted problems raised by a clear lack of diversity at the highest levels of health management. Around 90% of NHS chief executives and chairs are white males, and only around 1 in 5 of the total workforce come from underserved communities [34], which is a clear lack of representation in the system, which reduces the true presentation of the issues that matter the most for this underrepresented part of the community. However, the population served at a particular location may be almost entirely from one underserved group [34]. Our findings were similar to that concluded by Neville et al., 2022 [35], who concluded that the participation of diverse groups in the healthcare population will require a commitment from providers to partner and facilitate inter-sectoral collaboration with members of those communities. Whilst it may require time to redress this fully as health professionals require many years of training and experience, accepting, changing attitudes, and inviting and accepting increased input from the target population should be more quickly achievable to improve the situation in a shorter timeframe.

This study included only one sample of 60 people, but the sample achieved a broad representation of interested groups.

## 11. Limitations

The main limitation of this study was limited funding, which imposed limitations on the number of participants who could be hosted and the length of the venue hire. Another limitation was the nature of the data collected due to the use of unstructured focus groups. Some participants used personal experience, which can identify them or their healthcare providers, and the data could not be de-identified or used.

## 12. Conclusions

There is work to do regarding cultural stigma, which is a barrier to accessing support/care, but it should not be used as a smokescreen to disguise the fact that many from underserved backgrounds do reach out to healthcare services and are simply not being seen/considered. This is the most urgent barrier to address. Participants indicated that cultural stigma is not the only reason people do not receive help. People do not receive help because either there are no services available in their area or, if there are, they are fully booked for very long periods, in some cases more than 12 months.

The data also indicate that, despite extensive discussions about underserved communities avoiding services, these services were often perceived as inappropriate because they were seldom designed in collaboration with community members. Consequently, participants advocated for increased co-production with the Integrated Care Board (ICB) and the introduction of more mental health champions within their communities to offer guidance and support during the waiting period for NHS services. Additionally, an asset-based community development approach would facilitate better representation and culturally appropriate support.

## Figures and Tables

**Figure 1 healthcare-12-02506-f001:**
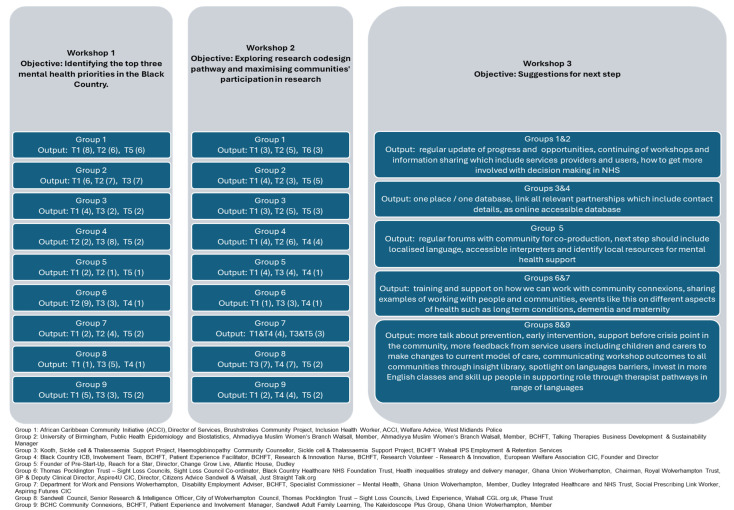
Themes and statements generated by the workshop groups.

**Figure 2 healthcare-12-02506-f002:**
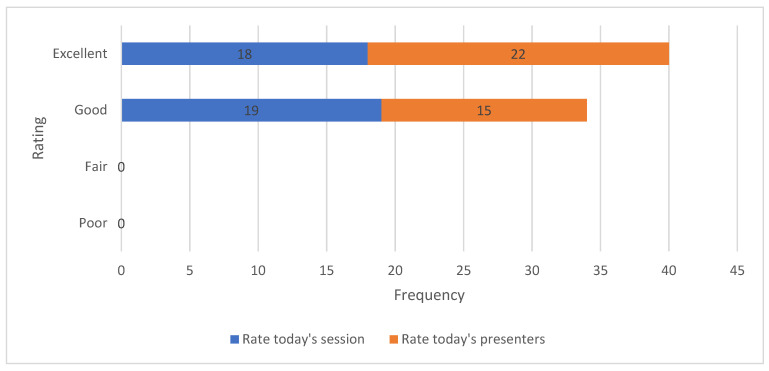
Event rating.

**Figure 3 healthcare-12-02506-f003:**
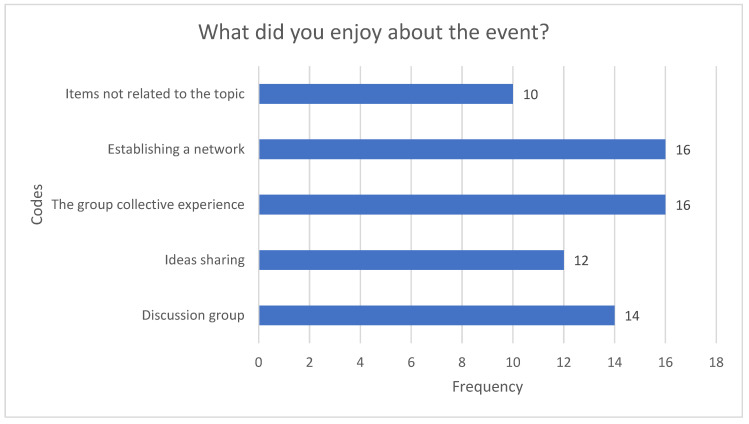
Question 3.

**Figure 4 healthcare-12-02506-f004:**
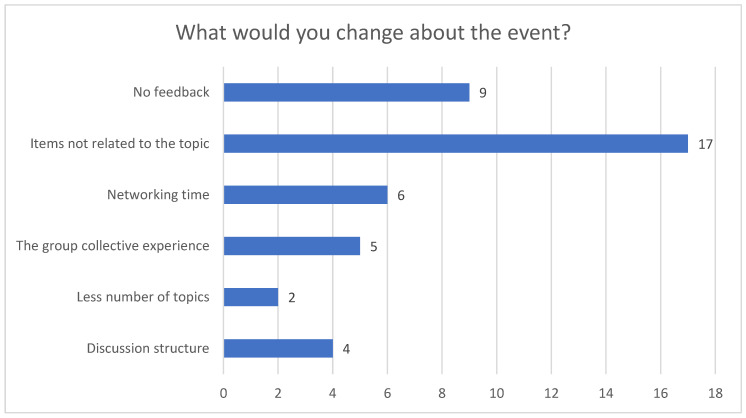
Question 4.

**Figure 5 healthcare-12-02506-f005:**
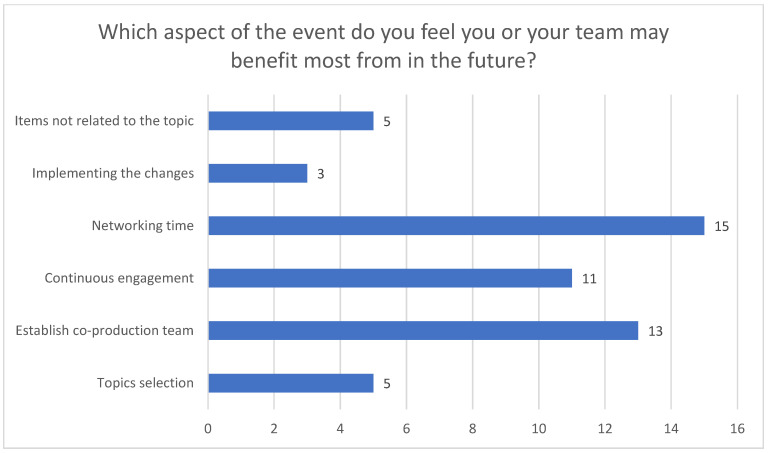
Question 5.

**Table 1 healthcare-12-02506-t001:** Generated codes based on the groups’ statements.

Codes Number	Codes Name	Frequencies	Percentage
Code 12	Ineligibility to access healthcare	17	1.98
Code 4	Few face-to-face appointments	22	2.56
Code 1	Few practices	34	3.96
Code 3	Large number of patients	37	4.31
Code 11	Poor self-care, self-worth or self-care capacity	44	5.12
Code 7	Poor patient experience	48	5.59
Code 5	Shift from on-site family GP-led care to generic GP care, generic allied health care, online appointments or self-care	66	7.68
Code 2	Limited appropriately trained workforce	82	9.55
Code 6	Change in patient expectations	109	12.69
Code 10	Poor communication between patients and healthcare professionals	116	13.50
Code 8	Complex healthcare needs	134	15.60
Code 9	Disparity due to language, culture, beliefs, age, gender and socioeconomic status	150	17.46

**Table 2 healthcare-12-02506-t002:** Themes based on the generated codes.

	Themes	Frequency	Percentage
Workshop 1		n = 116	
1	Barriers to appropriate healthcare	29	25%
2	Inappropriate prioritisation of accessible funded healthcare needs	25	22%
3	Limited resources for training, health promotion, preventative care and support networks	36	31%
4	New health services model	7	6%
5	Poor access to needed care	19	16%
Workshop 2		n = 117	
1	Research on barriers and enablers to access the needed care	24	20.5%
2	Research on the appropriate prioritisation of accessible funded healthcare need	17	14.5%
3	Research on the needed resources for training, health promotion, preventative care and support networks	26	22%
4	Research on a new health services model	28	24%
5	Research for patient benefit, not researchers, healthcare professionals or government	22	19%

## Data Availability

Data are contained within the article.

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
