# Peer review of "Identifying the Black Country’s Top Mental Health Research Priorities Using a Collaborative Workshop Approach: Community Connexions"

_healthcare, 2024, doi:10.3390/healthcare12242506_

Round 1
Reviewer 1 Report
Comments and Suggestions for Authors
Clear and engaging abstract and introduction.
When reporting statistics in 61-69, it would be helpful to indicate if the percentage referred to is the national average or the statistic for that area, and in any case, provide both for clarity.
Clear aims and objectives set out—this is very helpful to the reader.
Clear, excellent discussion of the methods. Again, very helpful to the reader and presented in a very engaging way.
Clear and helpful visuals on the coding process.
The discussion is clearly presented and engaging.
The conclusion is clearly written and provides a good finishing commentary on the discussion and research.
Author Response
When reporting statistics in 61-69, it would be helpful to indicate if the percentage referred to is the national average or the statistic for that area, and in any case, provide both for clarity. - Corrected Clear aims and objectives set out—this is very helpful to the reader. Clear, excellent discussion of the methods. Again, very helpful to the reader and presented in a very engaging way. Clear and helpful visuals on the coding process. The discussion is clearly presented and engaging. The conclusion is clearly written and provides a good finishing commentary on the discussion and research. - Thank youReviewer 2 Report
Comments and Suggestions for Authors
This is a great topic to study. I was pleased with the historical explanation of Black country and the amazing desire to use research to improve services for Black country residents.
Although the study is admirable by its foci, it lacks in many essential areas which are critical to determine accuracy, rigor, checks and balances, and credibility of qualitative data.
First, the introduction should include a paragraph with detailed information on mental health outcomes in Black country as well as specific stats on how COVID affected services in terms of presentation, type of concerns, etc. More information is also needed I guess more information on the cultural diversity of this area is needed on prior studies of Black country residents' perception of mental health, mental health utilization, mental health treatment, diversity and cultural humility of mental health professionals, and reasons for precocious termination---so, some information on mental health disparities across various social identities in that area.
About the Methods section
Also, are missing the philosophical underpinnings of the study as well as the type of mixed method used including the qualitative research method employed and how it best aligns with the theory, aims/objectives.
A demographic table is needed as well as the data analytic plan (we do not know what qualitative research method was used and therefore how the data analysis was conducted for the qualitative portion and if accuracy, rigor, checks and balances, and credibility factors were employed). This information will allow the reviewer to provide a more comprehensive review of the sections results and discussion.

Author Response
This is a great topic to study. I was pleased with the historical explanation of Black country and the amazing desire to use research to improve services for Black country residents. - Thank you Although the study is admirable by its foci, it lacks in many essential areas which are critical to determine accuracy, rigor, checks and balances, and credibility of qualitative data. First, the introduction should include a paragraph with detailed information on mental health outcomes in Black country as well as specific stats on how COVID affected services in terms of presentation, type of concerns, etc. - Corrected however, the relationship to COVID was not part of this effort More information is also needed I guess more information on the cultural diversity of this area is needed on prior studies of Black country residents' perception of mental health, mental health utilization, mental health treatment, diversity and cultural humility of mental health professionals, and reasons for precocious termination---so, some information on mental health disparities across various social identities in that area. - The researchers are aware of similar studies having been undertaken in the city of Birmingham (cited) and the North Midlands. It is precisely this information that the present study was attempting to inform. About the Methods section Also, are missing the philosophical underpinnings of the study as well as the type of mixed method used to include the qualitative research method employed and how it best aligns with the theory, aims/objectives. - Information added A demographic table is needed as well as the data analytic plan (we do not know what qualitative research method was used and therefore how the data analysis was conducted for the qualitative portion and if accuracy, rigor, checks and balances, and credibility factors were employed). This information will allow the reviewer to provide a more comprehensive review of the sections results and discussion. - All the data included in table 1, and figure 1Reviewer 3 Report
Comments and Suggestions for Authors
An important piece of work, well done. This paper can be strengthened with another proof read for better flow. I would also like to see a researcher positionality piece in there so I know the researchers relationship to the community and how that may affect the data that was collected. Please also see references below that will strengthen this article. Alofa atu.
Enari, D. (2021). Methodology marriage: Merging Western and Pacific research design.
Fa’alogo-Lilo, C., & Cartwright, C. (2021). Barriers and supports experienced by Pacific peoples in Aotearoa New Zealand’s mental health services. Journal of Cross-cultural psychology, 52(8-9), 752-770.
Neville, S., Wrapson, W., Savila, F., Napier, S., Paterson, J., Dewes, O., ... & Tautolo, E. S. (2022). Barriers to older Pacific peoples’ participation in the health-care system in Aotearoa New Zealand. Journal of Primary Health Care, 14(2), 124-129.
Author Response
An important piece of work, well done. This paper can be strengthened with another proof read for better flow. I would also like to see a researcher positionality piece in there so I know the researcher’s relationship to the community and how that may affect the data that was collected. Please also see references below that will strengthen this article.
- Thank you we used one of those recommended at the end of the discussion:
Our findings were similar to that concluded by Neville et al 2022, who concluded that participation of diverse groups in the population in health care will require a commitment from the providers to partnership and inter-sectoral collaboration with members of those communities.
Neville, S., Wrapson, W., Savila, F., Napier, S., Paterson, J., Dewes, O., ... & Tautolo, E. S. (2022). Barriers to older Pacific peoples’ participation in the health-care system in Aotearoa New Zealand. Journal of Primary Health Care, 14(2), 124-129.
Reviewer 4 Report
Comments and Suggestions for Authors
Lot of thanks for give me the chance to review a very interesting paper, such as “Identifying the Black Country’s top mental health research priorities using a collaborative workshops approach: Community 3 Connexions”. I would like to make some comments and suggestions to improve the chances for publication:
- The structure of the paper is not really academic and the writing style is correct, I appreciate the work of the authors. Perhaps the authors can modify the index, and include Aims and Objectives in other blocks (such a 1.2, 2.1...)
- Introduction is really good, with a lot of useful information.
- Main problem of the paper is the Method´s procedure. There is no explanation about the method (seems to be Grounded Theory, but it could be CDA), and especially the categories of the study, and they are fundamental in the qualitative part. Please, explain how the researchers have identified these categories and what they were.
- Results can be presented in APA tables, to make easier the reading and understanding.
- Discussion is very brief and there are no connections with similar studies. Please, rewrite this part and include more references.
- There is no part of Limitations and biases, and it´s also a basic part in every study, especially in a qualitative study.
- I highly recommend to include more references, especially more actual bibliography.
Author Response
Lot of thanks for give me the chance to review a very interesting paper, such as “Identifying the Black Country’s top mental health research priorities using a collaborative workshops approach: Community 3 Connexions”.
- Thank you
I would like to make some comments and suggestions to improve the chances for publication:
- The structure of the paper is not really academic and the writing style is correct, I appreciate the work of the authors. Perhaps the authors can modify the index, and include Aims and Objectives in other blocks (such a 1.2, 2.1...)
- Thank you, however we followed the journal template which does not require numbered paragraphs. Happy to number the sub-sections if the journal allows
Introduction is really good, with a lot of useful information.
- Thank you
Main problem of the paper is the Method´s procedure. There is no explanation about the method (seems to be Grounded Theory, but it could be CDA), and especially the categories of the study, and they are fundamental in the qualitative part. Please, explain how the researchers have identified these categories and what they were.
- Amended
Results can be presented in APA tables, to make easier the reading and understanding.
- Due to the nature of the results, this was the only style we were able to adopt.
Discussion is very brief and there are no connections with similar studies. Please, rewrite this part and include more references.
- We added new references into the discussion
There is no part of Limitations and biases, and it´s also a basic part in every study, especially in a qualitative study.
- Added
I highly recommend including more references, especially more actual bibliography.
- Five new references were added.